# The impact of clergy sexual abuse on spirituality and health: A systematic scoping review of the literature

Joanne Durkin[1]*, Rachel Zordan[2,3], Matthew Bullen[4‡], Nadia Pavich[1‡],
Patricia Therese Benedict Thomas[5‡], Carolyn Lethborg[6,7], Wendy Holder[8‡],
Melinda Jolly[9‡], Darlene Dreise[1‡], Daniel Fleming[1,10]

1 St Vincent's Health Australia, Melbourne, Victoria, Australia, 2 St Vincent's Hospital Melbourne, Melbourne, Victoria, Australia, 3 School of Health University of Melbourne, Melbourne, Victoria, Australia, 4 Private Consultancy, Mountain Stillness Listening Spaces, Sydney, New South Wales, Australia, 5 Grief Care, Catholic Cemeteries and Crematoria, Sydney, New South Wales, Australia, 6 Health Equity Research, St Vincent's Hospital Melbourne, Melbourne, Victoria, Australia, 7 Centre for Rural Health, College of Health & Medicine, University of Tasmania, Launceston, Tasmania, Australia, 8 Private Consultancy, Wendy Holder Psychologist, Launceston, Tasmania, Australia, 9 Pastoral Care St Vincent's Private Hospital Sydney, Sydney, New South Wales, Australia, 10 School of Medicine, The University of Notre Dame Australia, Melbourne, Victoria, Australia

☉ These authors contributed equally to this work.
‡ These authors also contributed equally to this work.
* joanne.durkin@svha.org.au

## Abstract

Sexual abuse perpetrated by clergy or other Church leaders can cause harm to the spirituality of people who are subject to this abuse, which also has impacts on their overall health. We conducted a systematic scoping review to examine how Spiritual Harm in this context is operationalized in the literature with specific reference to the Catholic Church. Literature searches were conducted across Academic Search Complete, Informit Database, ProQuest, Web of Science and Google Scholar in 2022 and updated throughout 2023. Eligible studies were published between 2002–2022 and included peer reviewed empirical research, systematic reviews, discussion, or perspective papers that explored spiritual and/or religious harm and sexual abuse in a church context, and the psychological, emotional, and spiritual impact of the abuse. The review included 12 research articles. Data were analysed using qualitative approaches and presented as a narrative summary. Reporting follows the Preferred Reporting Items for Systematic Reviews and Meta-Analyses guidelines for scoping reviews. Spiritual Harm in people who are subject to clerical sexual abuse is understood as a distinct condition or phenomena, comprising of complex spiritual, emotional and psychological components. However, it is not consistently defined. Understanding Spiritual Harm in a more comprehensive and consistent way is important in order to be able to adequately respond to the needs of victims, survivors, their families and wider communities.

**Data availability statement:** All relevant data are within the paper and its Supporting information files.

**Funding:** This project is funded by the Health Equity Grant St Vincent's Health Australia. We confirm that the funders had no role in study design, data collection and analysis, decision to publish, or preparation of the manuscript.

**Competing interests:** The authors have declared that no competing interests exist.

## 1. Introduction

There are long term psychological, emotional and physical health consequences for people who have experienced sexual abuse [1–3]. These consequences are related to the perpetration of harm against the person and to post-trauma stress associated with the abuse [4]. As such, this kind of abuse is often a source of enduring trauma, having a profoundly negative impact on the person, their self-identity [5], their family and wider community [4,6] which can endure across a lifetime [7].

In addition to these well-established consequences of sexual abuse, when such abuse is perpetrated by clergy or other religious leaders it is often referred to as a form of spiritual abuse [8] which causes a Spiritual Harm, adding to the complex nature of resulting trauma. Such harm is related to but distinguishable from the well-established psychological, emotional and physical health consequences noted above. Furthermore, some argue that this Spiritual Harm is further compounded when, upon disclosing clerical sexual abuse, a survivor is not believed, or treated poorly as a result of their disclosure, which has been a consistent feature of responses to abuse in Catholic Church contexts [1,6,8,9]. Whilst abuse is widespread across many religious communities [10,11], in this review we focus on the experience within the Catholic Church (hereafter, the Church) in particular. This focus reflects the wide scale of abuse in the Church [12] and its unique social and theological context [13], which a number of commentators suggest are contributing factors both to the occurrence of abuse and the trauma that follows it [14].

Even though there is agreement among commentators regarding the damage to a person's spirituality [8,15] when sexual abuse is perpetrated by a religious authority, the term 'Spiritual Harm' (and other synonyms used in the literature) is beset with a lack of conceptual clarity. We argue that addressing this gap is necessary to move research in this area forward: without clarity in defining Spiritual Harm, it is difficult to create responses orientated towards healing [16]. Correlatively, accurately describing harm in cases of complex trauma creates conditions of possibility for genuinely therapeutic responses [17].

In view of the current limitations in the literature and an emerging notion that addressing the trauma of survivors of clerical sexual abuse involves addressing their spirituality [18–20], and the prevalence of and particularity of this harm within the Catholic Church [14] we determined that a systematic scoping review of the literature was needed to understand how the term and associated concepts were being operationalized in contemporary academic literature.

## 2. Background and focus of the review

This study was initiated as part of a broader project in which was commissioned by a health care service in Australia. The central goal of the project is addressing the harm experienced by survivors of sexual abuse in the Church within health and spiritual care contexts. The project is supported by the Trauma Informed Spiritual Care Initiative Advisory (hereafter, the Advisory) which is comprised of clinicians, experts in trauma informed care, and individuals with lived experience of sexual abuse in the Church.

In the early stages of the project, the Advisory's engagement with literature revealed a lack of conceptual clarity in relation to spiritual harm when related to sexual abuse by religious authorities, particularly in a Church setting. This literature review formalizes our attention to that specific gap. Given this, our focus is narrowly defined as the spiritual harm that survivors experience when they have been subjected to sexual abuse in Church contexts. We recognize that spiritual harm can occur in other contexts (e.g., in familial contexts). We also recognize that there are many spiritual, psychological and emotional characteristics present for sexual

assault survivors who are not abused by clergy and/or Church leaders. Notwithstanding that these are important areas for study in their own right, they are outside the scope of the present study.

## 3. The current review

We conducted a systematic scoping review of existing literature describing Spiritual Harm in the context of sexual abuse in a Church setting. The aim of this scoping review was to examine its nature and associated spiritual, psychological and emotional characteristics of the term. The review had two specific research questions: (1) how is this Spiritual Harm operationalized in the available literature? and; (2) what are the key characteristics related to this Spiritual Harm?

## 4. Positionality

The research team comprised members of the Advisory. JD is the project officer and health researcher, DF is the project lead, a theologian and ethicist. Professional backgrounds of the authors include psychology practice and research (WH and RZ), pastoral and/or spiritual care (MB, PT, NP, MJ), social work practice and research (CL), and Aboriginal and/or Torres Strait Islander health, wellbeing and Reconciliation programs (MJ and DD). Some authors have lived experienced of sexual abuse in a Church context as survivors and some support survivors of sexual assault through their research, and/or advocacy for victims and survivors but have not experienced this harm themselves. All authors have experience of Catholicism either through their current or past experiences as members of the Catholic community, their spiritual practices, or their experiences in employment or education. Throughout this review, all authors had open dialogue about their professional, personal and spiritual experiences and considered the influence this had on understanding and decisions regarding the reporting of results. These conversations helped us to understand potential biases and blind spots in relation to the scoping review. Collectively, the authors agreed to adhere to the original protocol, clearly and transparently document all processes undertaken in order to minimise the potential for undue influence based in personal perspectives. The authors took every precaution to mitigate against personal perspectives having an undue influence on the reporting of results.

## 5. Methods

We designed and tested a protocol for a systematic scoping review [21] prior to commencing the study. This determined that the methodology was appropriate to understand how a concept was conceptualized, defined and operationalized [22]. The protocol for this review was based on the approach that Coffey et al. [23] used to investigate the term 'eco-anxiety' which similarly sought to answer questions related to operationalization and key characteristics of a term related to subjective experience. The present systematic scoping review is reported as per the Preferred Reporting Items for Systematic Reviews and Meta-Analyses guidelines for scoping reviews (PRISMA-ScR) [21] (See S1 File. PRISMA-SCR Checklist).

## 6. Search strategy development and data sources

The search was conducted in three stages (1) initial test searching of relevant databases to analyze search terms and key words, (2) keywords identified from the initial test searches used in a revised search strategy in all databases, and (3) reference lists of key returns mined for suitable keywords. The initial test searches were conducted August-September 2022 in ProQuest

and Academic Search Complete. This initial test search identified that various terms are used to discuss the spiritual component to the harm caused by sexual abuse in a religious context. These range from, but are not limited, to spiritual harm, spiritual damage, spiritual wounding, spiritual trauma, spiritual disconnection, dissociation, spiritual rape, and soul murder. The search strategy was further developed in consultation with the Advisory to ensure key concepts were captured. The search strategy was finalized in November 2022 and conducted in Academic Search Complete, Informit Database, ProQuest and Web of Science. An adapted search was also conducted in Google Scholar and the first 100 returns were screened. The search terms included: (spiritual or religion) AND (harm or abuse or trauma) AND (clerical OR clergy OR church OR religio* OR cathol*). English language restrictions were applied to the databases with a search period from January 2002 to November 2022. Initially, we sought contemporary literature in a 10-year timeframe. Due to the limited number of papers available, this was extended to 20 years.

The final searches were completed on the 22 November 2022. A sample search string with limiters is provided in Table 1. A full list of all database searches with exact search strings, limiters, and action taken is included in S2 File. Database Searches.

## 7. Search outcomes

A search of databases identified n = 631 articles of which 201 were identified as duplicates through an automated process using Endnote X9 [24] and manual identification by JD. The remaining n = 430 articles were reviewed for relevance and a further n = 368 were deemed unrelated in any way to the topic under investigation and removed by JD after discussion with co-authors. The process of title and abstract screening of the n = 430 articles was conducted in Endnote X9 initially, before 62 articles were imported into COVIDENCE [25] where two authors (JD and RZ) conducted title and abstract screening independently. Of the records (n = 62) screened, n = 23 were excluded at title and abstract screening. The remaining n = 39 studies were independently assessed by two authors (JD and DF) against the full exclusion and exclusion criteria outlined in Table 2. In total, n = 12 articles met the inclusion criteria and were included in the review. Handsearching of the references in the 12 included articles yielded no further papers. The process is documented in the PRISMA flow Chart (see Fig 1). Throughout 2023, the authors continued to update the searches to identify any

**Table 1. Example of a search string.**

| |
|---|
| **Search String:** spiritual* OR religio* OR devout OR spiritual* OR religio* OR devout; harm OR abuse OR trauma* OR damage OR abuse OR trauma* AND clerical OR clergy OR church OR religio* OR cathol*) AND AB (clerical OR clergy OR church OR religio* OR cathol* |
| **Limiters** Date 2002 – 2022; English Language; |

**Table 2. Inclusion and exclusion criteria.**

| Inclusion Criteria | Exclusion Criteria |
|---|---|
| ▪ Empirical research; systematic reviews; discussion papers, perspective papers<br>▪ Quantitative, observational, qualitative and mixed methods studies<br>▪ Spiritual and/or religious harm is examined in the context of sexual abuse in a Catholic Church context<br>▪ Article discusses trauma, mental health, and related outcomes<br>▪ Contains concept of psychological and/or emotional state when related to Spiritual Harm<br>▪ Published between January 2002 – October 2022<br>▪ English Language | ▪ Spiritual and/or religious harm in the context of sexual abuse by religious authority in the Church is not:<br>  ◦ specifically addressed, or;<br>  ◦ addressed as a particular harm, or;<br>  ◦ is not the focus of the article/paper.<br>▪ Letters to the editor; book chapters or reviews, trade publications; periodicals, news items, conference abstracts<br>▪ No explicit empirical, theoretical, or practical focus in relation to Spiritual Harm and sexual abuse or associated concepts. |

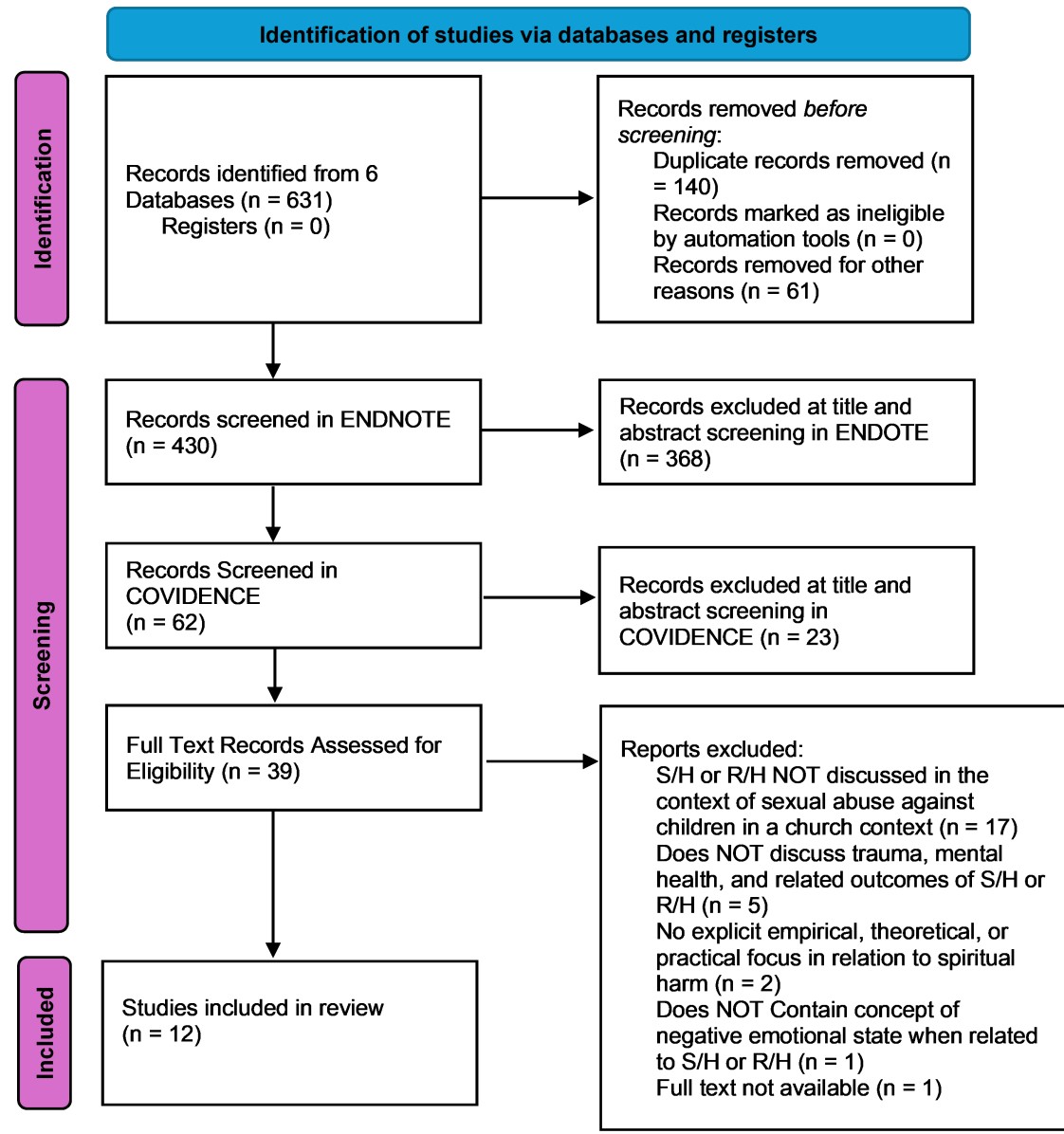

Note: S/H = Spiritual Harm; R/H = Religious Harm

**Fig 1. PRISMA flow chart.**

additional papers that may be included in the review that may have been published between November 2022 and November 2023. No additional papers were identified or included in this review.

## 8. Data charting; extraction, analysis and reporting of results

The first author (JD) imported n = 12 articles into NVIVO v12 Plus software [26] and data extracted under 'Nodes' that mapped to the extraction form developed in the study protocol. Extraction fields included author/year, publication type, sample (if appropriate), definitions

of Spiritual Harm and/or alternative descriptors of Spiritual Harm, key components as described in the literature, psychological and emotional components associated with Spiritual Harm, sample characteristics, measures used and recommendations and/or calls for further research. Author JD completed initial extraction of data in NVIVO. A presentation of this extracted data was given by JD to the Advisory in February 2023, after which all authors were sent a summary table of the full extraction along with a link to the full paper for review. Each author reviewed 2–3 papers independently and provided comment on salient points. The key points identified by the Advisory were compared against the extracted findings to ensure that the Advisory's contribution had been captured in full. Qualitative data were then consolidated into categories in order to ensure the research aims were fulfilled [27]. Categories included how the term was being operationalized in the literature then what the spiritual, emotional and psychological components of spiritual harm. Further categorization related to descriptors of the impact of the harm, conceptual notes and information relating to the act of abuse or perpetration of spiritual harm was also captured and presented. A summary of this data is presented in S3 File. Operationalising SH in the Literature. To illustrate the frequency of terms used to describe Spiritual Harm and associated concepts and provide a visual representation of important terms and key concepts, all extracted data was imported into https://worditout.com/ to create a word cloud (See Fig 2). Quality appraisal was not undertaken which is consistent current methodological guidelines related to conducting scoping reviews [21,22,28].

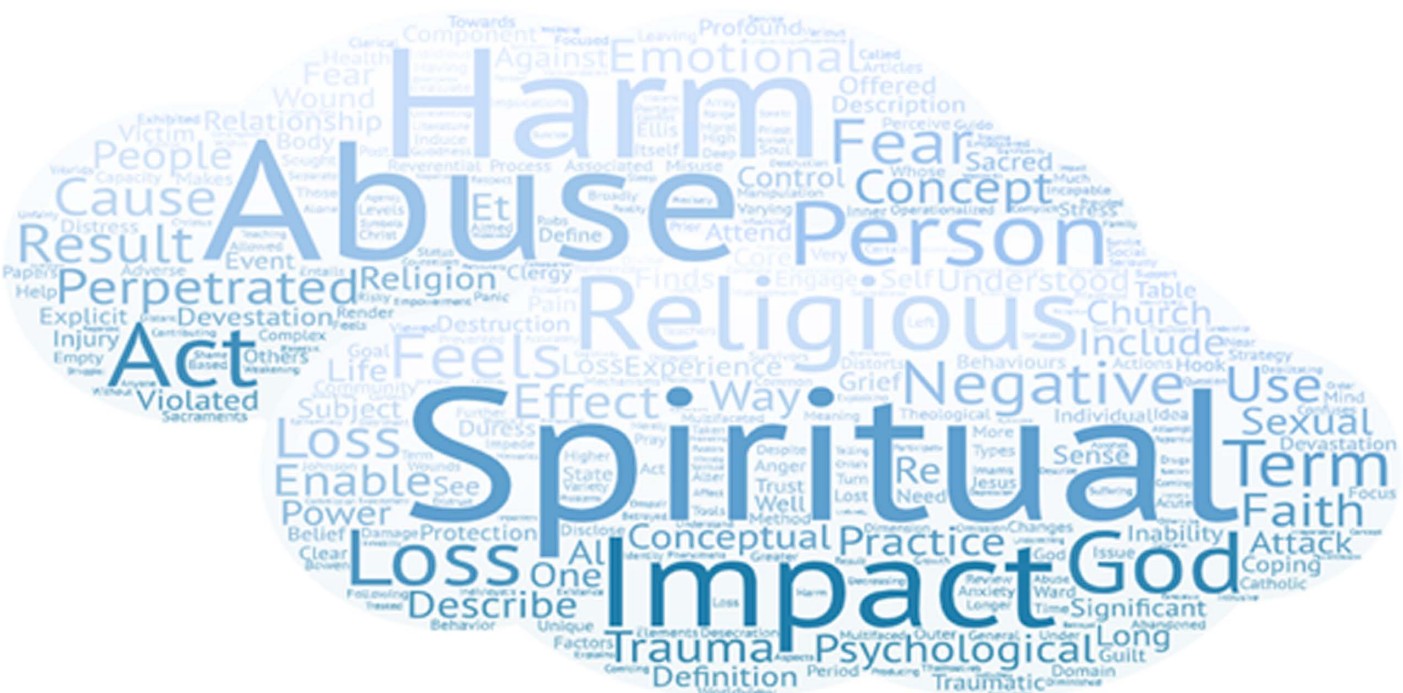

**Fig 2. Word cloud highlighting the broad range of vocabulary and phrases in the included literature to illustrate the various concepts relating to Spiritual Harm.**

## 9. Results of the review

### 9.1. Characteristics of included studies

Table 3 presents a summary of key characteristics of the studies included in this scoping review. Of 12 included studies, 5 were discursive and/or position pieces [8,15,29–31], with two position papers based on analysis of secondary documents [32,33]. Empirical studies comprised cross-sectional quantitative study (n = 1) [34], mixed method analysis of secondary documents (n = 1) [35], systematic literature review (n = 1) [36] and a thesis using quantitative methods (n = 1) [37]. One study interviewed people who have firsthand lived experience of abuse by Roman Catholic priests or religious [38]. The total participant sample in all included empirical studies was n = 123.

The majority of authors were from the United States of America (n = 7) [8,15,29,30,35–37], followed by Australia (n = 2) [32,33], the United Kingdom (n = 1) [38]; and further papers originating from Spain (n = 1) [34], and Slovenia (n = 1) [31]. None of the included papers examined Spiritual Harm and sexual abuse against children by religious authority from an Indigenous perspective.

### 9.2. Operationalization of Spiritual Harm in the literature

Within the reviewed literature, Spiritual Harm is described or understood in terms of the method or way in which the abuse was perpetrated against a person, contributing factors that enabled the abuse to be perpetrated and/or the effect on a person experience following the abuse. Authors variably describe the concept as a Spiritual Harm [32], a Spiritual Trauma [8,33], Religious Distress [29], Spiritual Injury [37], Spiritual Trauma [8], Religious/Spiritual Abuse and Trauma [36], Theological Trauma [38], and Spiritual Damage [34]. Further articles focused specifically on the Spiritual Dimension of Clergy Sexual Abuse [15] and the concept of Religious Duress and Reverential Fear that empowered the abuse [35].

In our review we found that the term Spiritual Harm and associated concepts are broadly operationalized as multifaced and complex. Common elements in the definitions or description of the term are the manipulation of religious practices to enable grooming, abuse and ongoing control of victims. Description of concepts include reference to the act of abuse (e.g., the use of religious practice or position to sexually, physically, psychologically and/or emotionally abuse another), the enablers that allowed the abuse to occur (e.g., power, position, and fear), the effect or impact of the resultant abuse perpetrated against the individual, the family and wider community (e.g., spiritual, emotional and psychological impact), the changes in behavior exhibited by people subjected to abuse (e.g., negative impact on person and relationship with God or their Church community) or the loss they experience. S3 File outlines the key terms used to operationalize Spiritual Harm in the extant literature.

We note that it was consistently the case that no explicit definition of 'Spiritual Harm' is offered by authors of included studies in this review. However, we found that papers provided clear descriptions of the act of spiritual abuse or the resulting impact the acts of harm had on a individuals' spirituality, religious attendance, and practice. Conceptually, Spiritual Harm was described in varying ways across studies [36] and was a complex concept [31]. Despite descriptions of the emotional and psychological harms associated with the concept, Spiritual Harm was understood to be a phenomenon in its own right [33]. Fig 2 provides a word cloud highlighting the broad range of vocabulary and phrases to illustrate the various concepts relating to Spiritual Harm.

In what follows, we distinguish between the way in which the concept is operationalized in terms of its spiritual, emotional and psychological features

**Table 3. Included studies with characteristics.**

| Author/ Year | Country | Title | Concept focus | Aims/ objective of paper/ study | Type of contribution/ research | Sample | Measures used |
|---|---|---|---|---|---|---|---|
| Benkert & Doyle [29] | **United States of America** | Clericalism, Religious Duress and its Psychological Impact on Victims of Clergy Sexual Abuse. | Religious duress | Discussion paper/ Aims not stated. | Discursive paper | n/a | |
| Bland [37] | **United States of America** | The psychological and spiritual effects of child sexual abuse when the perpetrator is a Catholic priest. | Spiritual Injury | To explore the negative, long-term psychological and spiritual effects of child sexual abuse when the perpetrator was a Catholic priest. | Thesis | Survivors n = 73 of childhood clerical sexual abuse | Religiosity Index, Spiritual Injury Scale; Trauma Symptom Checklist-40 (TSC-40); Sexual Abuse History and Healing Questionnaire. |
| Doyle [8] | **United States of America** | The Spiritual Trauma Experienced by Victims of Sexual Abuse by Catholic Clergy. | Spiritual Trauma | This article examines some of the symptoms and possible recovery from the spiritual dimension of post abuse trauma. | Discursive paper | n/a | |
| Ellis et al., [36] | **United States of America** | Religious/spiritual abuse and trauma: A systematic review of the empirical literature. | Religious/ Spiritual Abuse and Trauma | To provide an overview of the extant empirical research that has been conducted on religious/spiritual abuse or trauma. | Literature review | n/a | |
| Farrell, [38] | **United Kingdom** | Sexual abuse perpetrated by Roman Catholic priests and religious. | Theological Trauma | The focus of the research was upon the unique, subjective experience of a participant's understanding and awareness, of encountering an abusive sexual experience perpetrated by either a cleric or a religious. | Data were collected via a face-to-face semi structured interview | n = 12 individuals who had been sexually abused by Roman Catholic priests or religious | (PTSD-QAI) (Farrell, Keenan, & Taylor, 2003) based upon the DSM-IV TR (2000) classification for PTSD. Idiosyncratic Trauma Questionnaire (ITQ) (Farrell & Taylor, 2003) |
| Guido, [30] | **United States of America** | A Unique Betrayal: Clergy Sexual Abuse in the Context of the Catholic Religious Tradition. | Spiritual and religious abuse | Discussion paper/aims not stated | Discursive paper | n/a | |
| McPhillips [32] | **Australia** | Silence, Secrecy and Power: Understanding the Royal Commission Findings into the Failure of Religious Organisations to Protect | Spiritual Harm | Provide background and context to the establishment of the Royal Commission, Overview of the Commission's methodology investigation Analyse the outcomes of that investigation | Discursive/position paper; Analysis of outcomes related to the Royal Commission (Australia) | n/a | |
| Novsak et al., [31] | **Slovenia** | Therapeutic implications of religious-related emotional abuse. | Religious related emotional abuse | To discuss the emotional repercussions of religious-related abuse in the family. | Discursive | n/a | |
| Pargament et al., [15] | **United States of America** | Problem and Solution: The Spiritual Dimension of Clergy Sexual Abuse and its Impact on Survivors. | Spiritual Dimension of Clergy Sexual Abuse | To understand the spiritual dimension of clergy-perpetrated sexual abuse. | Discursive | n/a | |
| Pereda & Segura [34] | **Spain** | Child Sexual Abuse Within the Roman Catholic Church in Spain: A Descriptive Study of Abuse Characteristics, Victims' Faith, and Spirituality. | Spiritual damage | To examine adult victims of child sexual abuse by representatives of the Spanish Catholic Church. | Cross sectional study design sample comprised | n = 38 adults | Questionnaire |

*(Continued)*

**Table 3.** (Continued)

| Author/ Year | Country | Title | Concept focus | Aims/ objective of paper/ study | Type of contribution/ research | Sample | Measures used |
|---|---|---|---|---|---|---|---|
| Spraitz & Bowen [35] | **United States of America** | Religious Duress and Reverential Fear in Clergy Sexual Abuse Cases: Examination of Victims' Reports and Recommendations for Change. | Religious Duress and Reverential Fear | The discussion focuses on why victims remain silent and provides recommendations for new policy and for improving existing policy. | Mixed method; secondary analysis of documents | n = 16/18 priest files. | Qualitative retrospective content analysis |
| McPhillips [33] | **Australia** | "Soul Murder": Investigating Spiritual Trauma at the Royal Commission | Spiritual Trauma | This article examines existing research on spiritual trauma with regard to child sexual abuse. | Discursive/position paper; Analysis of secondary source | | Applies a five-point classification model developed by Kenneth Pargament and colleagues [15] for identifying and analysing spiritual damage. |

## 9.3. Spiritual components and impact of Spiritual Harm

In our review we found that Spiritual Harm is understood as a devastation of the spiritual aspects of a person [8,29,33] that has long-term negative spiritual effects [8,37]. These effects include feeling distant from or cut-off completely from God [33,36], being in conflict with God [38], a belief that God has violated [29,30,33] or abandoned the individual who has been abused [8,29], and/or a sense that God treated them unfairly by not preventing their suffering [37]. Sexual abuse in this context is often framed as being commissioned by a representative of God and thereby understood as an attack on the soul of the person [8,29], an attack on the core self or identity [33,36] and an attack on a person's sacramental worldview [30,33]. Guido [30] explains the particularity of this harm in the context of the Church: "Precisely because the priest is regarded by Catholics as an alter Christus, another Christ, his violation of a child's or adolescent's body is also a violation of a sacred trust and worldview" (p. 255). Spiritual Harm can also result in moral confusion [29] and considerable theological and existential struggles [8].The individual may feel that the central tools for making sense of themselves and their world that faith provided has been taken from them [8,30,33,38]. Spiritual harm is also described as a deep despair at the loss of relationship with God, often a foundational element of a religious person's experience of the world and their understanding of their purpose, value, vocation and communal identity [39], leaving the person isolated and alone as their relationship with God is ruptured and irreparable [8]. This can confiscate any spiritual coping mechanisms a person may have possessed individually and/or through community prior to their abuse [37].

The abuse also has dramatic consequences for religious observance [33,37] causing a distrust of religion and religious communities [33,36]. A person may experience an inability to pray, attend religious service and/or engage in the sacraments [8,30,32,33,38] which in turn robs these central elements of religious practice of their attendant meaning [30]. While individuals who have experienced Spiritual Harm are sometimes drawn to engage or re-engage with religious practice [33,36], they can be re-traumatized if they disclose abuse to family and religious community, particularly if the response is not supportive, compounding the Spiritual Harm and leaving them empty, lost and fearful [8,38].

### 9.4. Psychological components of Spiritual Harm

Our review also found that Spiritual Harm is bound together with psychological implications [8] which include an array of post-traumatic stress symptoms [33] such as anxiety and depression [8,15,36,37], panic attacks and other anxiety reactions, particularly when a survivor is in proximity to a church [29], sleep disturbances [30,37], sexual problems [37], and numbness and immobility which distorts their sense of reality [29] and a debilitating loss of independent functional status [15].

The psychological impact of Spiritual Harm is identified as causing indescribable pain that may confuse and psychologically overwhelm the individual, rendering them incapable of processing the sexual abuse [8,29]. Furthermore, it has serious impacts on a person's agency, wellbeing [32] and long-term health outcomes [33]. Often, survivors of abuse were prevented from disclosing their abuse for a prolonged period because of fear of repercussions, guilt or shame [8,29]. Left without coping strategies and protective mechanisms otherwise available to them [8,29,37], survivors of abuse may engage in maladaptive coping strategies to survive [36]. These can include avoidance and dissociative behaviours [30], risky sexual behaviours [15,36], substance use and misuse [15], self-injurious behaviours [36] or suicidality [15,33].

### 9.5. Emotional components of Spiritual Harm

Our review also found that Spiritual Harm is understood as having significant adverse emotional components [36] that can be devastating and cause profound distress [8,32,38]. Feelings of deep anger towards their abuser and the Church are core emotional components [8,29,30,33,38] which have a significant negative impact on the person. Individuals may experience feelings of toxic guilt and immobilizing fear, both because of the abuse and their inability to disclose what happened to them [8,29,35,37]. Spiritual Harm is also understood as bound together with an emotional trauma and betrayal [33,36], emotional manipulation, or emotional pain, which can render the person incapable of protecting and promoting their own emotional growth and spiritual well-being [8,29]. Feeling a loss of trust in institutions and others is also reported [15,32,33,36] as are feelings of abandonment [8,38] and an acute and debilitating emptiness [8,29]. As outlined by Doyle [8]:

> "Those who have been sexually assaulted by Catholic clergy or religious have experienced spiritual trauma as well as emotional and psychological trauma. The impact on the soul is often subtle and grows more painful and debilitating as time passes. Many survivors have said that this spiritual pain has been worse than the emotional pain." (p. 240)

### 9.6. Measurement tools

Measurement tools used in the investigation into Spiritual Harm and associated concepts included the Spiritual Injury Scale [37]; Religiosity Index [37]; Sexual abuse History and Healing Questionnaire [37]. Other measurement tools included the use of The Post-Traumatic Stress Disorder QAI (PTSD-QAI) [38,40] and Idiosyncratic Trauma Questionnaire (ITQ) [38].

### 9.7. Identified gaps and further research directions

Within the studies included in this scoping review, we extracted both calls for further research and the authors' suggestions for improvements and changes to support offered to people who have experienced Spiritual Harm and abuse in the Church. Authors identified a need for

further research into the psychological and spiritual functioning of people who had been sexually abused in a religious context [34,36,37], noting specifically the need to understand unique spiritual context within which the abuse occurred [34,38]. There were calls for comparison of outcomes based on gender of the person who experienced abuse [34,37], as well as calls to understand the difference between clerical or religious-related sexual abuse and other types of abuse, e.g., familial abuse [36]. The need to improve the measures used to assess Religious and Spiritual Abuse and Harm and to develop alternative measures to understand Religious and Spiritual Abuse and Harm was identified [36]. The importance of improving the diagnosis of such harm and understanding its impact on psychological and spiritual functioning for people who experienced clerical sexual abuse was also identified [38], as was the need to understand whether spiritual healing can aid psychological and emotional healing [37].

Studies included in this review found that care givers responding to Spiritual Harm need to understand the sacramental culture of Catholicism [30,33] in order to more comprehensively understand the harm. Further, there were calls for care givers to attend to the spiritual dimension of the harm and consider spirituality as a possible source for healing as well as the source of the harm [15,33]. Some authors proposed that redress processes for survivors of sexual abuse by clergy need to include spiritual restitution and recovery of those affected [32,33], with calls for religious organisations to collaborate in the spiritual healing process for survivors [33,34]. It was noted that initiatives sponsored by the Church tend not to explore the effect of abuse on survivors of sexual violence in a Church context, nor include substantial efforts to find ways to provide effective assistance and healing [8,33].

## 10. Discussion

The aim of this scoping review was to examine the literature describing Spiritual Harm in the context of sexual abuse in a Church context, to understand how the term is operationalized, and understand the key characteristics used the describe the concept. The key findings from our scoping review reveal a lack of clarity in both operationalization and conceptualization of Spiritual Harm, particularly in relation to sexual abuse in the Church [41,42]. The review identified that a range of terms overlay Spiritual Harm with trauma, distress, injury, and damage. We found various interpretations of the harm itself being primarily religious or spiritual. Conceptually, there was a lack of clarity around the use of the term as a descriptor of a form of abuse, a form of harm, the enablers that allowed abuse or harm to be perpetrated, and/or the resultant effect or impact of the changes or loss a person experienced as a result of being subjected to the abuse. In light of this, we note that concepts that are left unexamined can inadvertently become substitutions for other terms [43], and lack of linguistic precision has a detrimental impact both on selection and operationalization of definitions of research, and the ability to compare empirical results [44–46] which prevent advancement of research in the field. The scoping review demonstrated that much of the literature includes discursive and/or position papers from those working with or close to people who have experienced Spiritual Harm. We acknowledge it is important to recognize the work of advocates who speak for the person who has been subjected to harm. That said, we suggest that if we are to advance understanding of Spiritual Harm, more research that focusses on first-person accounts are critical and researchers in this field must continue to focus on listening to and partnering with survivors to create opportunities for research.

Further key findings from our scoping review found that Spiritual Harm can be understood as binding together spiritual, emotional and psychological components. We found a limited number of measures for assessing the Spiritual Harm caused by sexual abuse in a Church context. The development of consistently applied methodologies to measure the emotional,

psychological and spiritual components of the Spiritual Harm are important if we are to advance our understanding of the concept. Furthermore, the use of validated scales of measurement are an important step in advancing any field [47]. Existing measures addressing the psychological and emotional components of Spiritual Harm, for example depression, anxiety, and suicidal ideation, may be useful. However, a robust understanding of the complexity and the context in which the harm occurred (e.g., within the Church by a representative of God) is critical to grasping Spiritual Harm more comprehensively for survivors of abuse within the Church. In other words, sexual abuse in the context of the Church creates a harm comprised of many features, which include the particularities of context. Spiritual Harm can cause a moral confusion and burden, intense and complex shame, culminating in a rupturing or severing of a person's relationship with their Spiritual self and/or God. Although not explicitly defined in the literature, this maps to the definition of a complex harm or trauma which is widely recognized in other fields.

Our scoping review highlights the need for clearer conceptualization of Spiritual Harm in the context of sexual abuse within the Church when conducting research. Through further research, we may better understand the complicated nature of Spiritual Harm and offer an appropriate remedy and treatment for individuals who, through no fault of their own, have borne an enormous and complex burden as a result of the abuse perpetrated against them.

## 11. Limitations and future research directions

This review was limited to peer-reviewed articles published in academic journals to allow for a focus on how this concept is understood in the contemporary literature. A review of books, grey literature, reports and evidence from other sources may yield important findings on the topic of Spiritual Harm and sexual abuse in a Church context. This review was also limited to papers published in English.

This review was limited to Spiritual Harm and associated concepts in the context of sexual abuse in a Church context. Further research could explore the concept of spiritual and/or religious harm in other contexts and yield useful results, particularly in comparison to Church contexts. The review identified that there is an opportunity for further research with victims and survivors of sexual abuse in other religious and/or faith settings. Such a review may uncover comparisons and contrasts between abuse suffered within the Catholic Church and other Christian denominations and other religions, as well as spiritual communities and cults, however this was not our particular focus.

Most studies included in this review drew on the perspectives of researchers and those working in the field who spoke on behalf of victims and survivors. We recognize these researchers were speaking on behalf of those who may be unable to tell their story. Research methodology needs to evolve to create safe spaces for victims and survivors if they are to be actively involved and their voices heard. Placing a trauma-informed lens on all aspects of research may facilitate the involvement of diverse, marginalized and/or underrepresented groups.

The lack of conceptual clarity on the concept of Spiritual Harm and/or abuse provides an opportunity for further research. Researchers are encouraged to provide clear definitions of terms, e.g., harm or abuse as direct action or resultant state of victim/survivor. We identified that there is a need for a therapeutic response focused on spiritual healing alongside psychological and emotional healing. Failure to acknowledge the spiritual component of the harm and/or abuse will not lead to a full understanding of the wounding victims and survivors have suffered. Clinicians and others in the helping professions need to recognize and treat disorders when present and a cause of significant morbidity or mortality. Given these gaps, Spiritual Harm is largely unrecognized, unaddressed, and therefore left untreated – as was the case

of Disorder Grief which was first identified in early 1990. It is now recognized as Prolonged Grief Disorder and included in the latest edition of the International Classification of Diseases (ICD-11) and Diagnostic and Statistical Manual of Mental Disorders (DSM-5) [48]. Our hope is that an advancement of understanding and conceptual clarity of Spiritual Harm could lead to similar developments in this area of profound need, and lead to a future in which survivors are able to gain more effective acknowledgement, treatment, and healing.

## 12. Conclusions

The scoping review found that Spiritual Harm is understood in the literature as a distinct condition or phenomena that is comprised of complex spiritual, emotional and psychological components. People who have experienced sexual abuse in a Church context often experience deep pain and ongoing suffering. The scoping review found that there is a need to develop our understanding of Spiritual Harm if we are to adequately respond to the needs of victims, survivors, their families and wider communities and provide genuine, holistic support. This review identified that treating a Spiritual Harm as a purely psychological and/or emotional harm is an incomplete response to the trauma carried by survivors and victims. While there are psychological and emotional components of this type of harm, these are bound together with the the spiritual harm experienced by survivors. Attending to only some aspects of harm is detrimental for the overall effectiveness of any response. As such, further research is essential to develop evidence-based strategies for responding to the needs of survivors and must address both the spiritual components of the harm and the particularity of the context within which the abuse occurred (. in this case, within the framework set by the theology, practice and authority structures of the Church).

If we are to respond to the needs of people who have been subjected to a Spiritual Harm as a result of sexual abuse in a Church context, we need to develop our understanding of this complicated harm.

## Supporting information

**S1 File. PRISMA-SCR checklist.**
(PDF)

**S2 File. Database searches.**
(DOCX)

**S3 File. Operationalising SH in the literature.**
(DOCX)

## Author contributions

**Conceptualization:** Joanne Durkin, Rachel Zordan, Matthew Bullen, Nadia Pavich, Patricia Therese Benedict Thomas, Carolyn Lethborg.

**Data curation:** Joanne Durkin, Rachel Zordan, Nadia Pavich, Patricia Therese Benedict Thomas, Wendy Holder, Daniel Fleming.

**Formal analysis:** Joanne Durkin, Rachel Zordan, Matthew Bullen, Nadia Pavich, Patricia Therese Benedict Thomas, Carolyn Lethborg, Wendy Holder, Melinda Jolly, Darlene Dreise, Daniel Fleming.

**Funding acquisition:** Daniel Fleming.

**Investigation:** Joanne Durkin, Matthew Bullen, Nadia Pavich, Patricia Therese Benedict Thomas, Daniel Fleming.

**Methodology:** Joanne Durkin.

**Project administration:** Joanne Durkin.

**Supervision:** Melinda Jolly.

**Validation:** Matthew Bullen, Nadia Pavich, Patricia Therese Benedict Thomas, Carolyn Lethborg, Wendy Holder, Melinda Jolly, Darlene Dreise, Daniel Fleming.

**Visualization:** Joanne Durkin, Matthew Bullen, Carolyn Lethborg, Wendy Holder, Melinda Jolly, Darlene Dreise, Daniel Fleming.

**Writing – original draft:** Joanne Durkin, Daniel Fleming.

**Writing – review & editing:** Joanne Durkin, Rachel Zordan, Matthew Bullen, Nadia Pavich, Patricia Therese Benedict Thomas, Carolyn Lethborg, Melinda Jolly, Darlene Dreise, Daniel Fleming.

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
