## [Decision Letter · Decision Letter 0]

8 Nov 2024

PONE-D-24-33887The impact of clergy sexual abuse on spirituality and health: a systematic scoping review of the literature.PLOS ONE

Dear Dr. Durkin,

Thank you for submitting your manuscript to PLOS ONE. After careful consideration, we feel that it has merit but does not fully meet PLOS ONE’s publication criteria as it currently stands. Therefore, we invite you to submit a revised version of the manuscript that addresses the points raised during the review process.

The reviewers gave the manuscript high marks and requested only minor revisions to the text. Therefore, I urge the authors to proceed by fulfilling all the requests as indicated in the comments.<!--EndFragment

We look forward to receiving your revised manuscript.

Kind regards,

Stefano Federici, Ph.D.

Academic Editor

PLOS ONE

Journal Requirements:

2. Peer review at PLOS ONE is not double-blinded (https://journals.plos.org/plosone/s/editorial-and-peer-review-process). For this reason, authors should include in the revised manuscript all the information removed for blind review.

“Health Equity Grant St Vincent's Health Australia “

4. We note that you have referenced (Unpublished Ph.D. Thesis) on page 25, which has currently not yet been accepted for publication. Please remove this from your References and amend this to state in the body of your manuscript: (ie “Bewick et al. [Unpublished]”) as detailed online in our guide for authors

Additional Editor Comments:

The reviewers gave the manuscript high marks and requested only minor revisions to the text. Therefore, I urge the authors to proceed by fulfilling all the requests as indicated in the comments.

Comments from the Editorial Office: Please justify your choice for focusing this study on the Catholic church.

Reviewers' comments:

Reviewer's Responses to Questions

**Comments to the Author**

1. Is the manuscript technically sound, and do the data support the conclusions?

Reviewer #1: Yes

Reviewer #2: Yes

Reviewer #3: Yes

Reviewer #4: Partly

2. Has the statistical analysis been performed appropriately and rigorously? 

Reviewer #1: Yes

Reviewer #2: Yes

Reviewer #3: N/A

Reviewer #4: Yes

3. Have the authors made all data underlying the findings in their manuscript fully available?

Reviewer #1: Yes

Reviewer #2: Yes

Reviewer #3: Yes

Reviewer #4: Yes

4. Is the manuscript presented in an intelligible fashion and written in standard English?

Reviewer #1: Yes

Reviewer #2: Yes

Reviewer #3: Yes

Reviewer #4: Yes

5. Review Comments to the Author

Reviewer #1: The author has provided an examination of scientific literature on the topic of spiritual harm, explaining the concept and its relevant measures. The paper is well designed, it indicates the complexity of the analyzed theoretical approach. However, the section DISCUSSION does not present well the main findings elaborated in this study. It has to be more detailed showing better the connections between the key concepts introduced in this study.

Reviewer #2: 1) the Author has well design study, it need to address more social behavior status also.

2) did Author observed this types of studies any journal which is investigating other factor or same?

2) did The Author try to find/compare economic factor and what cause of this sexual abuse?

Reviewer #3: Dear authors,

This is an excellent and rigorous systematic scoping review. It was very clear and easy to follow the steps that you had taken in understanding how Spiritual Harm is understood in contemporary academic journals. You do well to recognise the limitations and offer future research direction. It is technically sound and the steps are clear to replicate.

My recommendation is to accept, but there are two points that I am not clear on relating to 4. Positionality. The first is to offer a better explanation for what "...some understand the harm perpetrated through their research, support and/or advocacy for victims and survivors" means. I was not clear on what this harm looks like through the research that you do.

The second, you discuss at the end that you had open dialogue about your own experiences which is very important and you considered the influence, but my question is what did this influence have on your own individual understandings but also the larger decisions made regarding the reporting of the results.

On p.16 I found a double comma. Under background: "Comprised by" could be "comprised of". Otherwise, it is very well written.

Reviewer #4: The present manuscript The impact of clergy sexual abuse on spirituality and health: a systematic scoping review of the literature shows and details a quality scientific research contemplating all the components to be considered in order to publish in this journal. The search strategies implemented to support the research are adequate and are carefully detailed in the development of this manuscript. The three stages executed as part of the strategies are convenient and allow achieving the proposed objectives, and the data clearly show a systematic review of studies on the subject.

The conclusions should be addressed with greater precision and correspond to the central objective of the project for which this research has taken place, which the authors of the manuscript state is to address the harm experienced by survivors of sexual abuse in the Church in contexts of health and spiritual care, in this sense a greater forcefulness is lacking in the conclusions which do not allow contributing relevant information to the project in question.

The manuscript has been written in English, it is adequate for the parameters of the journal, the language used is simple and at the same time respects the parameters of spelling and writing.

6. PLOS authors have the option to publish the peer review history of their article (what does this mean? ). If published, this will include your full peer review and any attached files.

**Do you want your identity to be public for this peer review?** For information about this choice, including consent withdrawal, please see our Privacy Policy .

Reviewer #1: No

Reviewer #2: No

Reviewer #3: No

Reviewer #4: **Yes: ** Andrea Stefanía Angulo Prado

---

## [Author Response · Author response to Decision Letter 0]

21 Nov 2024

Dear Editorial Team and Reviewers,

Thank you for taking the time to consider our paper. I have provided a detailed list of every comment and our response. We have addressed all concerns and agree with all of the feedback and comments you provided. You have supported us in improving this important work and we appreciate the time and effort you took while reviewing our work. We have copied the table below. For the full list of comments and amendments, see the uploaded file Response to Reviewers.

Regards, Authors.

List of review comments and responses

Editorial and Peer Review Comments Response and Change

Thank you for stating the following financial disclosure:

“Health Equity Grant St Vincent's Health Australia “

• Thank you. This is useful. The section has been added to the funding section. The funders had no role in study design, data collection and analysis, decision to publish, or preparation of the manuscript

We note that you have referenced (Unpublished Ph.D. Thesis) on page 25, which has currently not yet been accepted for publication. Please remove this from your References and amend this to state in the body of your manuscript: (ie “Bewick et al. [Unpublished]”) as detailed online in our guide for authors

• Thank you. I have amended the reference and now listed two citations – one for the paper which lists the use of the PTSD-QAI (Farrell D. Sexual abuse perpetrated by Roman Catholic priests and religious. Mental Health, Religion & Culture. 2009;12(1):39-53.) and the other which is the unpublished thesis. This now states that the thesis is unpublished in the reference list rather than refer to a specific date.

Please include captions for your Supporting Information files at the end of your manuscript, and update any in-text citations to match accordingly. Please see our Supporting Information guidelines for more information: http://journals.plos.org/plosone/s/supporting-information.

• Thank you. All summary files have titles and have been listed at the end of the manuscript as instructed.

Comments from the Editorial Office: Please justify your choice for focusing this study on the Catholic church.

• Thank you, we have strengthened this justification in the introduction and the background and focus of the review by both adding citations to justify the focus and also adding additional text which highlights the particularity of the harm in the Catholic Church context. These changes are highlighted in the tracked change manuscript.

1. Is the manuscript technically sound, and do the data support the conclusions?

Reviewer #1: The author has provided an examination of scientific literature on the topic of spiritual harm, explaining the concept and its relevant measures. The paper is well designed, it indicates the complexity of the analyzed theoretical approach. Thank you.

However, the section DISCUSSION does not present well the main findings elaborated in this study.

It has to be more detailed showing better the connections between the key concepts introduced in this study.

• Thank you. We have been through the discussion and made these points clearer. The discussion now better connects to the key concepts in the study.

Reviewer #3: Dear authors,

This is an excellent and rigorous systematic scoping review. It was very clear and easy to follow the steps that you had taken in understanding how Spiritual Harm is understood in contemporary academic journals. You do well to recognise the limitations and offer future research direction. It is technically sound and the steps are clear to replicate.

My recommendation is to accept, but there are two points that I am not clear on relating to 4. Positionality. The first is to offer a better explanation for what "...some understand the harm perpetrated through their research, support and/or advocacy for victims and survivors" means. I was not clear on what this harm looks like through the research that you do.

• Thank you. We agree this wasn’t clear and have amended to read as: Some authors have lived experienced of sexual abuse in a Church context as survivors and some do not have a lived experience of the harm, but support survivors of sexual assault through their research, and/or advocacy for victims and survivors.

The second, you discuss at the end that you had open dialogue about your own experiences which is very important and you considered the influence, but my question is what did this influence have on your own individual understandings but also the larger decisions made regarding the reporting of the results.

• Thank you. We have strengthened this section and revised as follows:

Throughout this review, all authors had open dialogue about their professional, personal and spiritual experiences and considered the influence this had on understanding and decisions regarding the reporting of results. These conversations helped us to understand potential biases and blind spots we could encounter prior to undertaking the scoping review. This allowed us to ensure all biases were managed and avoided. Collectively, the authors agreed to adhere to the original protocol, clearly and transparently document all processes undertaken in order to minimise the potential for undue influence based in personal perspectives. The authors took every precaution to ensure that personal perspectives did not have an undue influence on the reporting of results.

On p.16 I found a double comma. Under background: "Comprised by" could be "comprised of". Otherwise, it is very well written.

• Comma removed and comprised of changed

Reviewer #4: The present manuscript The impact of clergy sexual abuse on spirituality and health: a systematic scoping review of the literature shows and details a quality scientific research contemplating all the components to be considered in order to publish in this journal.

The search strategies implemented to support the research are adequate and are carefully detailed in the development of this manuscript. The three stages executed as part of the strategies are convenient and allow achieving the proposed objectives, and the data clearly show a systematic review of studies on the subject.

• Thank you this is useful.

The conclusions should be addressed with greater precision and correspond to the central objective of the project for which this research has taken place, which the authors of the manuscript state is to address the harm experienced by survivors of sexual abuse in the Church in contexts of health and spiritual care, in this sense a greater forcefulness is lacking in the conclusions which do not allow contributing relevant information to the project in question.

• Thank you. We have revised the discussion and strengthened this section to make the conclusions clearer. We appreciate this feedback.

The manuscript has been written in English, it is adequate for the parameters of the journal, the language used is simple and at the same time respects the parameters of spelling and writing.

• Thank you

---

## [Decision Letter · Decision Letter 1]

6 Jan 2025

The impact of clergy sexual abuse on spirituality and health: a systematic scoping review of the literature.

PONE-D-24-33887R1

Dear Dr. Durkin,

We’re pleased to inform you that your manuscript has been judged scientifically suitable for publication and will be formally accepted for publication once it meets all outstanding technical requirements.

Kind regards,

Stefano Federici, Ph.D.

Academic Editor

PLOS ONE

Additional Editor Comments (optional):

Reviewers' comments:

Reviewer's Responses to Questions

**Comments to the Author**

1. If the authors have adequately addressed your comments raised in a previous round of review and you feel that this manuscript is now acceptable for publication, you may indicate that here to bypass the “Comments to the Author” section, enter your conflict of interest statement in the “Confidential to Editor” section, and submit your "Accept" recommendation.

Reviewer #2: All comments have been addressed

Reviewer #3: All comments have been addressed

2. Is the manuscript technically sound, and do the data support the conclusions?

Reviewer #2: Yes

Reviewer #3: Yes

3. Has the statistical analysis been performed appropriately and rigorously? 

Reviewer #2: Yes

Reviewer #3: N/A

4. Have the authors made all data underlying the findings in their manuscript fully available?

Reviewer #2: Yes

Reviewer #3: Yes

5. Is the manuscript presented in an intelligible fashion and written in standard English?

Reviewer #2: Yes

Reviewer #3: Yes

6. Review Comments to the Author

Reviewer #2: (No Response)

Reviewer #3: Revisions have been made to a high standard and addressed the concerns of the reviewers. Recommend accept.

7. PLOS authors have the option to publish the peer review history of their article (what does this mean? ). If published, this will include your full peer review and any attached files.

**Do you want your identity to be public for this peer review?** For information about this choice, including consent withdrawal, please see our Privacy Policy .

Reviewer #2: **Yes: ** Dr. Bhagwat Alapure

Reviewer #3: No

---

## [Editor Report · Acceptance letter]

PONE-D-24-33887R1

PLOS ONE

Dear Dr. Durkin,

I'm pleased to inform you that your manuscript has been deemed suitable for publication in PLOS ONE. Congratulations! Your manuscript is now being handed over to our production team.

Kind regards,

on behalf of

Prof. Stefano Federici

Academic Editor

PLOS ONE